# The Impact of a Low-Carbohydrate Diet on Micronutrient Intake and Status in Adolescents with Type 1 Diabetes

**DOI:** 10.3390/nu15061418

**Published:** 2023-03-15

**Authors:** Neriya Levran, Noah Levek, Bruria Sher, Noah Gruber, Arnon Afek, Efrat Monsonego-Ornan, Orit Pinhas-Hamiel

**Affiliations:** 1Paediatric Endocrine and Diabetes Unit, Edmond and Lily Safra Children’s Hospital, Chaim Sheba Medical Centre, Ramat-Gan 52621, Israel; 2National Juvenile Diabetes Centre, Maccabi Health Care Services, Ra’anana 4345020, Israel; 3The Institute of Biochemistry, Food Science and Nutrition, The Faculty of Agriculture, Food and Environment, The Hebrew University of Jerusalem, Rehovot 5290002, Israel; 4Devision of Nutrition Unit, Chaim Sheba Medical Centre, Ramat-Gan 5262000, Israel; 5Sackler Faculty of Medicine, Tel Aviv University, Tel Aviv 6997801, Israel; 6General Management, The Chaim Sheba Medical Centre, Tel Hashomer, Ramat-Gan 52621, Israel

**Keywords:** type 1 diabetes, vitamins, minerals, Food Frequency Questionnaire, HbA1c, low-carbohydrate diet

## Abstract

Objective: The aim of this study was to evaluate the macronutrient and micronutrient intake and status in youth with type 1 diabetes mellitus (T1DM) following the consumption of a low-carbohydrate diet (LCD). Research Methods and Procedures: In a prospective intervention clinical trial, adolescents with T1DM using a continuous glucose monitoring device were enrolled. Following a cooking workshop, each participant received a personalized diet regime based on LCD (50–80 g carbohydrate/day). A Food Frequency Questionnaire was administered, and laboratory tests were taken before and 6 months following the intervention. Twenty participants were enrolled. Results: The median age was 17 years (15; 19), and the median diabetes duration was 10 years (8; 12). During the six-months intervention, carbohydrate intake decreased from 266 g (204; 316) to 87 g (68; 95) (*p* = 0.004). Energy intake, the energy percent from ultra-processed food, and fiber intake decreased (*p* = 0.001, *p* = 0.024, and *p* < 0.0001, respectively). These changes were accompanied by declines in BMI z-score (*p* = 0.019) and waist-circumference percentile (*p* = 0.007). Improvement was observed in the median HbA1c from 8.1% (7.5; 9.4) to 7.7% (6.9; 8.2) (*p* = 0.021). Significant declines below the DRI were shown in median intake levels of iron, calcium, vitamin B1, and folate. Conclusions: The LCD lowered ultra-processed food consumption, BMI z-scores and the indices of central obesity. However, LCDs require close nutritional monitoring due to the possibility of nutrient deficiencies.

## 1. Introduction

Type 1 diabetes mellitus (T1DM) is one of the most common endocrine and metabolic conditions in childhood and is often characterized by poor glycemic control [1]. Integral to the management of T1DM, dietary therapy aims to provide healthy eating principles, improve diabetes outcomes, and reduce cardiovascular risk [2]. A Mediterranean diet in conjunction with carbohydrate counting is frequently used in the clinical management of T1DM [2]. Notwithstanding the pharmaceutical, technological, and nutritional advancements in the past decades, glycemic parameters remain suboptimal, with an average HbA1c of 9.3% between the ages of 15 and 18 years [3].

Carbohydrates are the primary macronutrient that affects the postprandial glycemic response. The international guidelines published by the International Society of Pediatric and Adolescent Diabetes include 40–50% of total energy consumption from carbohydrates, and achievement of optimal postprandial glycemic control with appropriately matched insulin to carbohydrate ratios and insulin delivery [2]. The evidence in the literature is currently insufficient to support the use of low-carbohydrate diets (LCD) as an adjunctive treatment for T1DM. However, given the difficulty of matching carbohydrate intake with insulin dose, reducing dietary carbohydrate consumption among people with diabetes has become a common dietary pattern [4,5].

The lower consumption of carbohydrates essentially lowers the glycemic response and the insulin requirement. Although LCD has a promising effect on glycemic control, endorsing the regime may lead to harmful dietary consequences. This is due to the complete or partial avoidance of healthy sources of carbohydrate foods such as whole grain bread, cereals, legumes, fruit, and vegetables.

In individuals with T1DM, adverse health risks such as diabetic ketoacidosis, hypoglycemia, dyslipidemia, glycogen depletion, and growth impairment remain clinical concerns. In an illustrative case series of children with T1DM who were on LCD, some experienced growth delay and fatigue [6]. In addition, adherence to restricted diets is challenging and can impact social normality.

Parents or people with T1DM who choose LCD usually do so without proper medical guidance and may put their child or themselves at risk of nutritional depletion of essential nutrients and minerals. Studies examining glycemic outcomes from LCD have largely been cross-sectional and without validated dietary data [7].

In the current study, we aimed to investigate the effect of an LCD intervention on macronutrient and micronutrient intake and status in youth with T1DM.

## 2. Materials and Methods

### 2.1. Participants and Study Design

This report documents a prospective intervention clinical trial conducted in the Pediatric Endocrinology and Diabetes Unit at the Sheba Medical Center. Eligibility criteria were a diagnosis of T1DM according to the American Diabetes Association criteria [1] for at least one year, age 12–22 years, and the usage of a continuous glucose monitoring device (Dexcom, San Diego, CA, USA, Medtronic, Northridge, CA, USA, Libre, Alameda, CA, USA). Exclusion criteria included a medical history of eating disorders in participants or their first-degree family members (there is a clear link between dieting and developing an eating disorder. We were concerned that a restrictive carbohydrate diet could result in, or aggravate, overeating and binge-eating behaviors in those at risk for eating disorders) or any other mental illness. Eligible trial participants were enrolled after they were contacted during their visits to the pediatric diabetic clinic. Written informed consent was obtained from participants aged 18 years or older and from parents or legal guardians of those under age 18 years. Ethics approval was obtained from the Helsinki Committee in the Sheba Medical Center.

### 2.2. Diet Intervention

At baseline, each participant underwent a cooking workshop and received a personalized diet regime based on the LCD. For participants younger than 18 years, nutrition education was provided to both the participants and their parents. Participants met individually with a dietitian for diet instructions and support at weeks 1, 2, 4, 7, 10, 12, and 24, for a total of seven frontal meetings. Twice during the first 12 weeks, the dietician conducted 10–15-min motivational telephone calls with each participant. During the entire course of the study, every participant had the option of consulting with the study’s dietitian (Appendix A).

### 2.3. Low-Carbohydrate Diet

The LCD aimed to provide 50–80 g/day of carbohydrates. There was no caloric restriction, but each patient received a weekly plan with main meals and snacks. The planned macronutrient composition of the diet (percentage of total calories) was: 20% carbohydrate, 25% protein, and 55% fat. All the dietary details were stated in the protocol and approved by the IRB.

### 2.4. Assessment of Nutritional Composition

The habitual food consumption of the participants was evaluated using the Food Frequency Questionnaire (FFQ), which was taken at baseline and after six months of intervention. The FFQ included 116 food items commonly eaten in Israel, standard portion sizes, and a frequency response section. It is based on a validated FFQ used for determining the dietary intake of Israeli multiethnic populations [8].

Using the Tzameret software, Israeli food and nutrient database, total energy intake (Kcal) and both macronutrients and micronutrients were calculated [9]. The distributions of macronutrients and micronutrients as percentages of daily energy consumption were also estimated and compared to dietary recommended intake (DRI) values [10].

### 2.5. Medical History and Anthropometric Measurements

Age of diabetes onset, diabetes duration, and other medical diagnoses data were retrieved from medical records. Height, weight, and waist circumference were measured at each visit according to standardized protocol by trained and certified staff. Body mass index (BMI) was calculated as weight (kg)/height squared (m^2^). BMI z-score norms were calculated for ages 2–20 years. For participants older than 20 years on the day of enrollment, we extrapolated the BMI z-score from the calculated BMI at age 20 [11].

### 2.6. Biochemical Parameters

Blood samples including HbA1c, total cholesterol, LDL cholesterol, and HDL cholesterol were collected under metabolic stability conditions. The latter were defined as no episode of diabetic ketoacidosis within 1 month before the visit and after ≥12 h of fasting. Laboratory results of serum C-reactive protein (CRP), blood urea nitrogen (BUN), creatinine, sodium, magnesium, calcium, zinc, phosphorous, vitamin B1, vitamin C, and folic acid were recorded. All fasting blood samples were taken at baseline and at 24 weeks, from a forearm vein and then processed by ELISA (Enzyme-Linked Immunosorbent Assay) at the Sheba Medical Center laboratories.

### 2.7. Trial Outcomes

Our primary endpoint was nutritional vitamins and mineral status after 24 weeks of an LCD. Secondary outcomes were body weight and waist circumference at this time point.

### 2.8. Statistical Analysis

Categorical variables were described using frequencies and percentages. Continuous variables were expressed as medians and interquartile ranges (IQR, 25th; 75th percentiles). The Wilcoxon test was used to compare continuous variables before and after the 6-month period. Spearman’s correlation coefficient test was used to study associations between continuous variables; >0.36 was considered as a moderate correlation, while r > 0.67 was considered as a high correlation. All statistical tests were two-sided, and all *p* values were adjusted by the false discovery rate. The statistical analyses were performed by SPSS software (IBM SPSS STATISTIC version 28, IBM Corp., Armonk, NY, USA, 2021).

## 3. Results

### 3.1. Study Group Characteristics

Twenty adolescents with T1DM (14 females) were enrolled in the study at median (IQR) age of 17 years (15; 19). The median diabetes duration was 10 years (8; 12). Eighteen participants were treated with an insulin pump and two were treated with multiple daily injections. The median BMI z-score was 1.3 (0.65; 1.50); nine participants were categorized as having normal weight, seven were categorized as having overweight, and four were categorized as having obesity. The median waist circumference was 85.7 cm (80.0; 91.8), and the median percentile was 76.5% (55.2; 83.5). One female participant withdrew after 3 months as she found it challenging to manage the LCD (Figure 1—A flow chart of the study).

### 3.2. FQQ

The median baseline percent from calories of carbohydrates was 44% (37; 47); from protein, 18% (16.5; 20); and from fat, 35% (30; 37). Baseline median percentages of micronutrients were calculated according to DRI values as follows: fiber 115% (97.5; 145.5), iron 101% (85; 138), magnesium 145% (118; 180), calcium 116% (82; 140.7), zinc 135% (111.2; 166.2), copper 158% (144; 182), vitamin B1 152% (136; 189), vitamin B2 229% (188; 264), vitamin B6 252% (199; 288), folate (vitamin B9) 123% (108; 143), vitamin B12 225% (200; 309), and vitamin C 359% (200; 471). After 6 months of the LCD intervention, the median intakes of several macronutrients and micronutrients were significantly different than at baseline (Table 1). The median percentage of calories from carbohydrate was 20% (18; 25); from protein, 25% (22; 25); and from fat, 51% (48–52). Median energy consumption decreased from 2537 kcal (1954; 2773) to 1533 kcal (1256; 1758 kcal) (*p* = 0.001), and the percent of calories from ultra-processed food decreased from 16.6% (9.4; 23.0) to 11.0% (8.3; 15.9) (*p* = 0.047). The reported carbohydrate intake decreased by 67%, from 265.0 g (204; 315) at baseline to 86.6 g (68.1; 95) (*p* < 0.001). Accordingly, this decline was seen in fibers, total sugars, and fructose. While the median protein intake decreased significantly, it remained within the DRI for all the participants. Fat intake, on the other hand, did not change significantly. Significant decreases were observed in the median intakes of several minerals and vitamins. Median intakes lower than recommended values were noted in iron, calcium, vitamin B1, and folate. Moderate decreases were observed in median intakes of fat-soluble vitamins (A, E, K) and in vitamin C, but these changes were not statistically significant (Table 1). According to the DRI, higher proportions of participants were deficient after than before the intervention, in: fiber, 75% vs. 35%; calcium, 50% vs. 45%; magnesium, 20% vs. 10%; copper 5% vs. 0%; vitamin B1, 65% vs. 15%; and folate, 50% vs. 20%.

### 3.3. Weight Loss and Waist Circumference

The LCD was associated with significant reductions in median BMI z-scores (*p* = 0.042) and waist circumference percentiles (*p* = 0.021) (Table 2).

### 3.4. Blood Laboratory Measurements

The median (interquartile range) HbA1c level declined after LCD, from 8.1% (7.5; 9.4) to 7.7% (6.9; 8.2), *p* = 0.021. Parameters of the lipid profile did not change significantly. CRP declined from 3.5 mg/L (1.1; 7.1) to 2.5 mg/L (1.0; 4.9) (*p* = 0.042). The median serum level of magnesium did not change; however, three participants had low levels (<1.8 mg/dL) at the end of the study. The median serum levels of folic acid and vitamin C did not change; however, one participant had borderline levels of folic acid deficiency (2–4 ng/mL), and another developed marginal vitamin C level (<6 mg/L). The median serum zinc level decreased from 130.5 mcg/dL (104.7; 150.0) to 98.0 mcg/dL (82.5; 119.0) (*p* = 0.042), but all participants were in the normal range of 50–150 mcg/dL. The medians of vitamin B1, calcium and phosphorous did not change significantly (Table 2).

### 3.5. Correlations

Delta body weight was not correlated with any of the parameters examined. The delta of calories from ultra-processed food did not correlate with any of the macronutrients or micronutrients examined. Table 3 shows correlations of decreased carbohydrate intake and of decreased protein intake, with changes in the consumption of selected macronutrients and micronutrients, as reported in the FFQ. A decreased intake of carbohydrates was associated with a significant decreased intake of fibers, iron, copper, potassium, magnesium, vitamin B1, vitamin B6, vitamin B2, and vitamin C. Decreased protein intake was significantly correlated with a decreased intake of fat, iron, calcium, potassium, sodium, zinc, vitamin B1, and vitamin B2; no correlations were observed between reported FFQ intakes and blood levels of calcium, magnesium, thiamine, and vitamin C.

## 4. Discussion

In this novel study of youth with TIDM who followed LCD for six months, decreases were found in median intakes of several macronutrients and micronutrients. Median blood levels of several nutrients decreased. These changes were in parallel to an improved median HbA1c level and lower median values of CRP, BMI z-score, and waist circumference.

The proportion of calories from ultra-processed food decreased during the intervention, because adherence to the LCD required more frequent cooking and less consumption of prepared food. Our findings are in concordance with a randomized trial among adults without diabetes [12].

The decrease by 70% in carbohydrate intake shows good compliance of our participants throughout the study period. We believe that the workshops with food preparation and imparting knowledge, together with the careful supervision and frequent in-person meetings and phone calls contributed to the good adherence and compliance in our study.

The decrease in carbohydrate intake was associated with reduced intakes of fibers and fructose. Low fiber intake was consistently demonstrated in several studies that investigated LCD [6,13,14]. Appropriate fiber intake has been positively linked with a potential increase in lifespan [15].

Decreased carbohydrate intake requires the substitution of another macronutrient, hence the fear that increased fat consumption could impair the lipid profile. However, among our participants, the fat intake did not change significantly after conversion to LCD due to calories decrease. Lipid profiles were not significantly elevated; these findings are in line with a 12-week crossover randomized study comparing high-carbohydrate diet versus LCD in 14 participants with T1DM [16]. Daily protein intake, on the other hand, decreased but was still higher than or as high as the DRI.

The reported FFQ showed significantly lower median intakes at the end of the intervention compared to baseline, according to percentages of recommended daily allowance of several minerals and vitamins (iron, calcium, phosphorus, vitamin B1). These findings are in agreement with a systematic review that showed deficient intakes of magnesium, calcium, iron, iodine, thiamine, and folate in healthy adults who followed a carbohydrate-restricted diet [17]. Only one case series examined micronutrient intake among adolescents and children with T1DM who consumed LCD [6]. Among five participants, deficient intakes were detected in at least one of the following; calcium, magnesium, phosphorus, thiamine, and folate [6]. These data suggest that some high-carbohydrate foods are good sources of vitamins and minerals, which are present naturally or added to fortify foods. At the end of the 6-month intervention, decreases by about 30% were reported in the median intakes of water-soluble vitamins, namely, vitamin B1 and folate. Our findings of unchanged median blood levels, despite decreases in median intakes, corroborate a prospective study that measured the nutrient intake of children and adolescents on 4-month treatment with a ketogenic diet [18]. We demonstrated that the decreases in mineral and vitamin intake were associated with decreased intake of carbohydrates and not secondary to the decrease in total energy. Indeed, the vitamins whose intake decreased are mainly found in whole grains, which are not part of an LCD; and in pork and seafood, which are not regular components of the Israeli diet. B vitamins are critical cofactors for axonal transport, the synthesis of neurotransmitters, and many cellular metabolic pathways. Thus, guidelines for nutritional replacement should be given to those who consume LCD [19]. Importantly, mean intakes of fat-soluble vitamins (A, E, K) did not change during our intervention. This might be due to the composition and types of food consumed in the LCD, such as eggs, fish, meat, and nuts.

Median intake levels of the trace elements iron and calcium were lower after the LCD than at baseline. Notably, a systematic review and case series described a lower intake of iron after conversion to LCD [6,17].

In the present study, median serum zinc values were within the normal range both at baseline and at the end of the intervention in spite of a downward trend. Data on the effect of dietary zinc intake on its serum level remain inconclusive. This decrease in daily consumption may be due to the lack of legumes and grains in LCD. Zinc is an essential trace element found in food, playing a role in many antioxidative defense and metabolic processes, such as insulin processing, storage, secretion and action [20,21]. Studies have shown that serum zinc concentrations of Polish children with T1DM were significantly lower than those of a control group [22]. One review estimated a dose–response relation of zinc intake with serum zinc concentrations in children aged 1–17 years [23], while another more recent study showed that serum zinc concentrations were not related to dietary or supplemental zinc but to an individual’s sex and age as well as the time of blood draw [24]. Homeostatic regulatory mechanisms were also observed to result in decreased dietary zinc absorption as intake increased [23], emphasizing the importance of monitoring zinc intake rather than relying solely on zinc levels in plasma.

Data indicated a decrease in copper intake. Copper is a vital mineral that aids in the production of neurotransmitters, connective tissue, neuropeptides, and energy [25]. Amounts of the mineral in food vary based on the season, soil quality, and water sources [26]. A case study found copper deficiency-induced anemia and neutropenia in a child who changed from a liquid to solid ketogenic diet [27].

According to the FFQ, magnesium intake was significantly lower in our participants after the LCD; four participants had deficient magnesium plasma levels. Magnesium is an essential cation that is found in legumes, nuts, and vegetables. Of note, magnesium deficiency is the most prevalent trace element deficiency in individuals with T1DM [22] and is associated with cardiac arrhythmias, hypertension, and intima-media thickness (an early marker of atherosclerosis) [28]. Therefore, individuals with T1DM who follow an LCD are in particular need of guidance for nutritional enrichment or supplements of magnesium.

A 30% decrease in selenium intake was observed in our cohort despite instructed supplementation with Brazil nuts. Selenium is an essential trace element known to influence various physiological processes, including energy homeostasis, through its redox functions. It is used to synthesize selenoproteins, which is a family of proteins with mainly antioxidant roles [29]. A study on children with intractable epilepsy on a ketogenic diet showed 20% selenium deficiency [30] and suggested screening for selenium deficiency when intake is lower than 75%. Thus, an appropriate selenium level should be obtained through the diet, and the consumption of selenium-rich foods, such as seafood, organ meats, and Brazil nuts, should be recommended.

Our data demonstrated improved parameters of the metabolic syndrome such as weight, BMI z-score, waist circumference, and CRP, as well as improved glycemic control, among adolescents with T1DM following a 6-month LCD intervention. The high CRP levels observed at baseline corroborate a study of children with T1DM who had higher levels of CRP than a control group; elevated CRP has been linked to coronary events [31]. Furthermore, the incidence of the metabolic syndrome among TIDM is growing, and it is a major problem, further increasing risks of morbidity and mortality [31]. Therefore, the impact of LCD on improving these parameters is of high importance.

For people with T1DM who follow LCD, we advise long-term and periodic monitoring of the nutrients whose intakes and blood levels decreased during the six months of our intervention. Nutritional deficiencies after LCD emphasize the need to provide nutritional substitutes that could fill the nutritional gaps [32].

Our study has some limitations. As with all dietary recall studies, FFQ interviews are subject to recall bias. However, this recall method is considered more reliable than a single-day recall [33]. Despite its strong research foundation, the FFQ probably contains omissions regarding precise amounts of nutrients. Another limitation is that in Israel as elsewhere, a large market of ketogenic/low-carb products has emerged as a result of the popularity of LCD in the general public. As some producers are local, the quantities of the dietary components may not always have been precise. This could result in underestimation and overestimation of the total caloric intake. In addition, our cohort comprised only 19 participants and does not inform the development of nutritional deficiencies following long-term adherence to LCD. Although physical activity is a major aspect of lifestyle change, the participants did not receive explicit instructions regarding their physical activity. Moreover, as the aim was nutritional intake and not weight change, physical activity was not assessed. The strengths of the study are its prospective follow-up. It is the first study to perform a 6-month intervention, intensive dietary surveillance for LCD, and assessment of a variety of micronutrients.

## 5. Conclusions

Medical nutrition therapy remains a cornerstone of diabetes care [2]. LCD has the potential of significantly improving glycemic control, preventing and treating overweight and obesity, and improving parameters of the metabolic syndrome such as central obesity. Nonetheless, LCD may result in nutritional deficiencies, as decreased carbohydrate consumption was positively correlated with decreases in vitamin and mineral intake that were not due to calorie reduction. Thus, these decreases resulted from the LCD content rather than from reduction in calories. As LCD may be accompanied by nutritional deficiencies, in individuals with type 1 diabetes who have opted to switch to LCD, we recommend an assessment of the contents of vitamins and minerals from the beginning. The dietician should plan a diet enriched with foods containing soluble vitamins (in particular folate and thiamin), selenium, magnesium, calcium, and iron. Furthermore, we recommend supplementation in a specific quantity, such as yeast extract, Brazil nuts, and Wolffia globose daily. Finally, we advise checking vitamin and mineral blood levels every six months and, if necessary, supplementing daily intake with vitamins and minerals.

## Figures and Tables

**Figure 1 nutrients-15-01418-f001:**
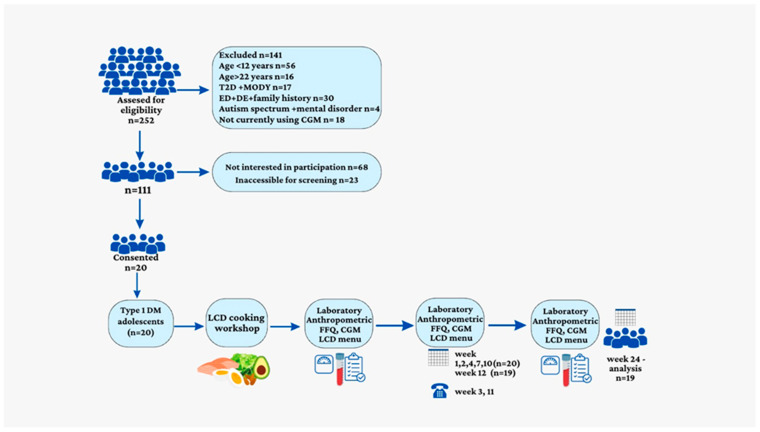
Flow chart of the study enrolment and regime.

**Table 1 nutrients-15-01418-t001:** Median (interquartile range) intakes of selected nutrients before and after a low-carbohydrate diet, according to the Food Frequency Questionnaire.

		At Baseline	After Six Months	Percent Change $\frac{after-before}{before}\times 100$	*p*-Value	Adjusted *p*-Value ^a^
Macronutrients	Carbohydrate, g	265 (204; 315)	86 (68; 95)	67.4	0.0001	<0.001
	Sugar alcohols, g	0.0 (0.0; 8.6)	0.0 (0.0; 2.5)	0	0.133	0.197
	Fiber, g	33 (28; 43)	21 (17; 28)	30.1	0.0001	<0.001
	Total sugars, g	120 (87; 156)	51 (42; 69)	56.9	0.0001	<0.001
	Fructose, g	25.8 (22.1; 36.0)	11.9 (6.7; 23.1)	53.9	0.0001	<0.001
	Protein, g	116 (92; 138)	91 (69; 108)	21.6	0.020	0.040
	Fat, g	85 (78; 108)	85 (65; 96)	0	0.445	0.531
	Cholesterol, mg	429 (369; 580)	420 (365; 617)	2	0.960	0.997
	Saturated fat, mg	25.0 (22.3; 34.5)	25.5 (19.4; 29.0)	2	0.460	0.536
Minerals	Iron, mg	15.6 (11.7; 17.1)	11.2 (8.8; 12.6)	28.2	0.001	0.004
	Magnesium, mg	508 (429; 604.	398 (361; 433)	21,6	0.002	0.007
	Phosphorus, mg	1793 (1615; 2323)	1462 (1181; 1637)	18.5	0.005	0.01
	Calcium, mg	1315 (902; 1468)	966 (635; 1232)	26.6	0.036	0.067
	Zinc, mg	12.6 (10.5; 15.8)	10.5 (8.3; 12.4)	16.7	0.016	0.034
	Copper, mg	2.1 (1.7; 2.3)	1.5 (1.3; 1.8)	23.8	0.001	0.004
	Selenium, mcg	160 (125; 206)	111 (89; 136)	30.1	0.010	0.022
	Potassium, mg	4803 (3763; 5768)	3605 (2990; 4316)	20	0.005	0.014
	Sodium, mg	4209 (3645; 5250)	3529 (2966; 4410)	16.1	0.070	0.122
Vitamins	Thiamine B1, mg	1.6 (1.5; 2.1)	0.9 (0.8; 1.2)	43.8	0.0001	<0.001
	Riboflavin B2, mg	2.6 (2.1; 3.0)	1.8 (1.4; 2.4)	31	0.001	0.004
	Niacin B3, mg	29.1 (22.6; 35.0)	24.7 (17.2; 28.4)	15	0.064	0.116
	Vitamin B6, mg	3.2 (2.6; 3.6)	2.1 (1.7; 2.6)	34.4	0.001	0.004
	Folate B9, mcg	492 (422; 602)	361 (286; 484)	26.5	0.004	0.01
	Vitamin B12, mcg	5.5 (4.7; 7.1)	5.8 (4.8; 7.7)	+5.4	0.904	0.984
	Vitamin C, mg	293 (156; 350)	212 (142; 309)	27.8	0.099	0.156
	Vitamin D, mcg	8.1 (6.3; 11.7)	8.7 (6.3; 10.4)	+7.4	0.930	0.990
	Carotene, mcg	7332 (5610; 10193)	6439 (3997; 8773)	12.2	0.126	0.984
	Vitamin A, mcg	1951 (1592; 3484)	1375 (1152; 2091)	29.6	0.084	0.141
	Vitamin E, mg	14.0 (11.2; 17.0)	12.7 (10.4; 15.8)	9.3	0.159	0.222

^a^ adjusted by the false discovery rate. Delta: after minus before; percent change: delta/before. Abbreviations: g: gram; mcg: microgram; mg: milligram.

**Table 2 nutrients-15-01418-t002:** Clinical and biochemical data of the participants before and after a low-carbohydrate diet intervention.

		Before	After	Change	*p*-Value	Adjusted *p*-Value *
Anthropometric measurements	BMI z-score	1.30 (0.65; −1.58)	1.20 (0.54; −1.49)	−0.13 (−0.29; −0.02)	0.008	0.042
	Waist circumference percentile	76.5 (55.5; 83.5)	65.5 (37.6; −77.5)	−6.5 (−19.0; −4.0)	0.002	0.021
Treatment	Insulin unit/kg	0.8 (0.63; −1.95)	0.7 (0.50; −0.92)	−0.12 (−0.18; −0.04)	0.006	0.01
Blood tests	HbA1c %	8.1 (7.5; 9.4)	7.7 (6.9; 8.2)	−0.8 (−1.3; −0.3)	0.001	0.021
	Total cholesterol mg/dL	176 (160; 194)	160 (155; 190)	0.00 (−22; 18)	0.289	0.379
	LDL cholesterol mg/dL	105 (97; 119)	110 (98; 120)	−3.5 (−14; 18)	0.737	0.814
	HDL cholesterol mg/dL	64 (53; 71)	58 (45; 65)	−11 (−4; 0)	0.02	0.06
	Triglycerides mg/dL	84 (64; 89)	69 (56; 88)	−8 (−20; 0)	0.019	0.06
	CRP <0.20–5.00 mg/L	3.5 (1.1; 7.1)	2.5 (1.0; 4.9)	−0.77 (−1.1; 0)	0.010	0.042
	Urea 17–45 mg/dL	28.0 (23.5; 33.25)	27.0 (21.0; 31.0)	−1.0 (−7.5; 0.5)	0.195	0.364
	Creatinine 0.62–1.10 mg/dL	0.63 (0.56; 0.7)	0.64 (0.56; 0.7)	0.0 (0.0; 0.0)	0.530	0.618
	Zinc 50.0–150.0 mcg/dL	130.5 (104.7; 150.0)	98.0 (82.5; 119.0)	−28 (−63; 0)	0.006	0.042
	Magnesium 1.90–2.70 mg/dL	1.9 (1.8; 2.0)	1.9 (1.8; 2.0)	0.0 (−0.2; 0)	0.260	0.364
	Phosphorus 2.00–4.00 mg/dL	4.1 (3.8; 4.4)	4.1 (3.7; 4.4)	0.0 (−0.2; 0.3)	1.000	>0.999
	Calcium 8.1–10.4 md/dL	9.7 (9.6; 10.1)	9.8 (9.5; 9.9)	0.1 (−0.4; 0.2)	0.260	0.364
	Vitamin b1 66.5–200.0 nmol/L	131.8 (119.4; 150.8)	129.0 (120.0; 36.9)	−4.2 (−15.4;−1.5)	0.260	0.364
	Vitamin C 4.6–14.9 mg/L	11.0 (8.0; 12.8)	11.8 (9.8; 14.4)	0.8 (−0.8; 3.8)	0.190	0.364
	Folic acid 5.9–24.0 ng/mL	9.9 (7.1; 13.5)	10.1 (7.4; 16.1)	0.6 (−1.8; 1.5)	0.230	0.364

The data are presented as medians and interquartile ranges. * Adjusted *p*-value by the false discovery rate. BMI: body mass index; HbA1c: glycated hemoglobin; CRP: C-reactive protein; IU/L: international units per liter; mg/L: milligrams per liter; mcg: microgram per deciliter; ng/mL: nanogram per milliliter.

**Table 3 nutrients-15-01418-t003:** Correlations of decreased carbohydrate intake and of decreased protein intake, with changes in consumption of selected macronutrients and micronutrients, as reported in the Food Frequency Questionnaire.

Correlation with Decreased Carbohydrate Intake (g)			
	R	*p*-Value	*p*-Value adj
Delta calories	0.749	<0.001	<0.001
Delta fiber	0.75	<0.001	<0.001
Delta sugar alcohols	0.180	0.447	0.448
Delta protein	0.341	0.141	0.172
Delta fat	0.412	0.079	0.108
Delta iron	0.598	0.005	0.011
Delta calcium	0.352	0.128	0.14
Delta copper	0.720	<0.001	<0.001
Delta potassium	0.653	0.002	0.005
Delta sodium	0.292	0.212	0.24
Delta magnesium	0.576	0.008	0.014
Delta zinc	0.37	0.108	0.132
Delta thiamine	0.701	0.001	0.002
Delta vitamin B6	0.538	0.014	0.002
Delta riboflavin	0.538	0.014	0.023
Delta vitamin C	0.675	0.001	0.002
**Correlations with Decreased Protein Intake (g)**			
	**R**	** *p* ** **-Value**	** *p* ** **-Value adj**
Delta calories	0.827	<0.0001	<0.0001
Delta fiber	0.214	0.366	0.402
Delta sugar alcohols	−0.039	0.869	0.869
Delta fat	0.826	<0.0001	<0.0001
Delta iron	0.627	0.003	0.005
Delta calcium	0.687	0.001	0.002
Delta copper	0.460	0.041	0.061
Delta potassium	0.595	0.006	0.011
Delta sodium	0.502	0.024	0.04
Delta magnesium	0.403	0.078	0.107
Delta zinc	0.947	<0.001	<0.0001
Delta thiamine	0.448	0.048	0.075
Delta vitamin B6	0.85	<0.0001	<0.0001
Delta riboflavin	0.850	<0.0001	<0.0001
Delta vitamin C	0.37	0.108	0.132

Delta: after minus before.

## Data Availability

The data and the questioners are all in Hebrew and could be sent by a personal request.

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
