# Peer review of "The Impact of a Low-Carbohydrate Diet on Micronutrient Intake and Status in Adolescents with Type 1 Diabetes"

_nutrients, 2023, doi:10.3390/nu15061418_

Round 1
Reviewer 1 Report
In this article, Levran et al. study the impact of a low-carbohydrate diet (LCD) in individuals with type 1 diabetes (T1D). In this 6-month intervention study with 20 participants, they observed significant improvement in the glycemic index as well as obesity-associated parameters after undertaking the LCD regime. However, they also observed a significant reduction in a few essential nutrients, including iron, calcium, vitamin b1, and folate. These findings suggest that there should be careful considerations with the LCD regime for individuals with T1D. The article is well-written, especially the discussion section is nicely elaborated. However, there are a few major concerns that limit the enthusiasm for this study in its current form. The authors need to thoroughly work on the data presentation to address these concerns:
- Firstly, the study lacks sufficient novelty. There are several recent publications on this very topic (e.g., PMID: 36364951, PMID: 34836158, PMID: 29735574, etc.) that have shown similar findings, especially with glycemic improvements.
- Secondly, the study lacks power. With a total of just 20 participants, that too categorized into different weight groups does not add to the confidence about the data. Although the authors mention this factor in their caveats, they need to address the issue experimentally.
- Finally, the current manuscript lacks critical information that makes the interpretation of the data difficult:
- Details about the suggested diet are missing. (What was the precise food that was suggested to the participants?) This is extremely important because that would clarify why there were reduced levels of certain nutrients. Factors like protein and fat intake, types of carbohydrates in the LCD regime, vegetarian vs. meat sources, etc., would expectedly impact the outcomes.
- The authors still need to provide the questionnaire they used for the participants to answer. Understanding the collected information and whether it covers all the aspects necessary for a clear interpretation would be essential.
- The authors also need to consider the aspect of physical activity of the participants as it would be an additional impacting factor.
Minor concerns/suggestions:
- A flow chart for the experimental regime that includes information about the total participants that started and how many were ultimately studied due to exclusion criteria would be easier to understand.
- Line 77: "Exclusion criteria included a medical history of eating disorders in participants or their first-degree family members, or any other mental illness." The authors need to explain the exclusion criteria as to why (physiological reasons) these participants were excluded.
- Line 79 is incomplete: "Eligible trial participants were enrolled after contacted during their visits to the."
Reviewer 2 Report
The authors have undertaken an interventive study on adolescents with type 1 diabetes with the main goal of understanding how a low carbohydrate diet (LCD) would reflect on the intake and serum levels of micronutrients. The study is interesting due to the increasing interest in these types of diets among persons with dysglycaemia. It also adds knowledge to the current literature, given that it was undertanken in a population that is not so well studied. The manuscript reads well and follows a logic rationale. I have some concerns, mainly regarding the study protocol in terms of the dietary intervention, which is not well described in the manuscript. Please see major and minor comments below.
Major comments:
- given that this was an intervention with clinical outcomes, has it been registered in any platform such as clinicaltrials.gov?
- In the methods, under diet intervention, there is no mention on what type of nutritional information was given to study participants;
- Also, what kind of recipes and recommendations were made to study participants that led to this large energy deficit?
-Did the ethical committee see the type of recommendations given to these individuals? Why aren't these shown in the manuscript? You can add them as an appendix if there is no space;
- I don't understand why did the authors not consider a normal BMI cathegory at baseline, considering that 9 participants had normal body weight/BMI;
- Result 3.5 correlates decreases of CHO and of protein intake with changes in micronutrients but the intake of protein was 25% En. Given that they ate about 1530kcal, this is about 95 or 96g of protein intake (authors report 91g after 6months), which seems to be normal. Then, how did it lead to micronutrient deficiency? Please comment or explain in the discussion;
- I would like to see how many of the study participants had a BMI z-score within the LOW range after the intervention. Given that 9 participants had normal body weight at baseline and that they reduced an average of 1000kcal/day (this is way far the recommended change in energy intake for this age group...), it is expected that some participants would have LOW body weight after the 6 months. Please add this info to the manuscript and comment it on the discussion section;
- I am also interested to know how did the authors handled this large energy deficit during the study. Knowing that participants were seen several times during the study, how did nutritionists deal with this dangerous energy deficit? Did you have a contingency plan for this? was this planned at all? If so, where is the reference that shows that it is safe to reduce 1000kcal in people of this age group??
- Did you have any information on Total daily dose of insulin? How did this change? This would be very important to show if available;
- Also, did the study participants report more hypoglycaemias than before the intervention? Did you access this? You did mention that they had continuous glucose monitoring, so I am assuming that this was monitored...
- Note that these safety aspects are critical for the study protocol and have to be considered before reproducing your study, so they must be transparently presented (especially the food and nutritional information given to study participants during the workshop and visits that led to the carbohydrate reduction)
Minor comments:
Abstract:
Line 27-28 must be rephrased; currently it reads: Improvements was observed median HbA1c 8.1% 27 (7.5;9.4) to 7.7% (6.9;8.2) (p=0.021). - Is it missing something? did you mean improvements were observed in HbA1c from median 8.1% to 7.7%?
Key words: missing Type 1 diabetes
Methods: Line 79 currently reads "(...) their visits to the." - the sentence is incomplete.
Results: table 1 must be formatted; the footnotes (a, b) are not shown anywhere in the table and the one about B12 and D (a) is not clear - did you mean the participants did not take supplements containing these vitamins?
- In table 2, the footnotes include: ; SGOT: serum glutamic-oxaloacetic transaminase; SGPT: serum glutamic-pyruvic transaminase; LDH: lactate dehydrogenase, but these parameters are not in the table. On the other hand, CRP is in the table but there is no footnote for it (C reactive protein).
Reviewer 3 Report
The aim of this prospective intervention clinical trial is to evaluate macro and micronutrient intake and status in T1DM youth following a LCD diet.
There are concerns related to LCD diets in T1DM subjects: 1. Caution should be applied as LCD showed a tendency toward more frequent hypoglycemia. 2. Reduced blood glucose variability, related to LCD, could have clinical relevance to individuals with T1DM 3. Excess (animal) protein and (animal) fat is a risk factor for chronic degenerative diseases that are obviously high in these patients.
There are important critical issues in this paper that prevent it from being published
1. the control group is absent, many of the changes reported by the authors as positive (e.g. decreasing BMI and decreasing HbA1c) are justified by calorie restriction
2. the sample is small
3. there is significant protein and calorie restriction which could explain all the benefits the authors attribute to carbohydrate restriction
4. there is a difference between the various types of carbohydrates. Fibre, starches and sugars are all carbohydrates but for the authors there is no difference.

Round 2
Reviewer 1 Report
The authors have addressed almost all of my concerns satisfactorily.
Author Response
Dear Reviewer,
Together with my collaborators, I would like to thank you.
Sincerely Neriya Levran and Orit Pinhas-HamielReviewer 2 Report
I am happy with the answers provided by the authors and also with the editions made to the manuscript
Author Response
Dear Reviewer,
Together with my collaborators, I would like to thank you.
Sincerely Neriya Levran and Orit Pinhas-Hamiel
Reviewer 3 Report
The authors agree with most of my observations.
Unfortunately, the paper is practically identical to the first version, only minor adjustments have been made.
Author Response
Dear Reviewer,
All together, from all three reviewers, we had 30 points that were altered according to the reviewers' suggestions and remarks. In addition, we added supplements to the diet plan and protocol (attached below) . We hope you will find the final version of the article suitable for publication.
